# Cardiovascular Manifestations in Multisystem Inflammatory Syndrome in Children (MIS-C) Associated with COVID-19 According to Age

**DOI:** 10.3390/children9050583

**Published:** 2022-04-20

**Authors:** Claudia Campanello, Claudia Mercuri, Maria Derchi, Gianluca Trocchio, Alessandro Consolaro, Roberta Caorsi, Angelo Ravelli, Alessandro Rimini, Maurizio Marasini, Marco Gattorno

**Affiliations:** 1Pediatrics and Neonatology Unit, San Paolo Hospital, 17100 Savona, Italy; 2Department of Neuroscience, Rehabilitation, Ophtalmology, Genetics, Maternal and Child Health (DINOGMI), IRCSS Istituto Giannina Gaslini, University of Genoa, 16147 Genoa, Italy; cla.mercuri0507@gmail.com (C.M.); alessandroconsolaro@gaslini.org (A.C.); angeloravelli@gaslini.org (A.R.); marcogattorno@gaslini.org (M.G.); 3Cardiology Unit, IRCSS Istituto Giannina Gaslini, 16147 Genoa, Italy; mariaderchi@gaslini.org (M.D.); gianlucatrocchio@gaslini.org (G.T.); alessandrorimini@gaslini.org (A.R.); mauriziomarasini@gaslini.org (M.M.); 4Pediatric Rheumatology Department, IRCSS Istituto Giannina Gaslini, 16147 Genoa, Italy; robertacaorsi@gaslini.org; 5Scientific Direction, IRCSS Istituto Giannina Gaslini, 16147 Genoa, Italy

**Keywords:** multisystem inflammatory syndrome in children, cardiovascular manifestations, pediatrics, COVID-19

## Abstract

Cardiac involvement in multisystem inflammatory syndrome in children (MIS-C) associated with coronavirus-19 disease is often observed with a high risk of heart failure. The aim is to describe cardiovascular involvement, management and early outcome in MIS-C by comparing cardiovascular manifestations in children younger and older than 6 years old. This retrospective observational study included 25 children with MIS-C, admitted to a single pediatric center between March 2020 and September 2021. The median age was 5 years (13 patients under 6 years and 12 over 6 years); coronary artery abnormalities were observed in 77% of preschoolers, with small and medium aneurysms in half of the cases and two cases of mild ventricular dysfunction. School-age children presented myopericardial involvement with mild to moderate ventricular dysfunction in 67% of cases, and two cases of transient coronary dilatation. There was a significant NT-pro-BNP and inflammatory markers increase in 25 of the patients, and mild elevation of troponin I in 9. All patients were treated with intravenous immunoglobulin and corticosteroids, and 8 with anakinra. None of the patients needed inotropes or intensive care unit admission. Our study shows the frequent cardiovascular involvement in MIS-C with a peculiar distribution, according to different age group: coronary artery anomalies were more frequent in the younger group, and myopericardial disease in the older one. A prompt multitarget, anti-inflammatory therapy could probably contribute to a favorable outcome.

## 1. Introduction

Multisystem inflammatory syndrome in children (MIS-C) is a rare complication of coronavirus disease 2019 (COVID-19), caused by a severe acute respiratory syndrome coronavirus 2 (SARS-CoV-2) infection. Although initial reports described less frequent and milder forms of COVID-19 in children than in adults [1], since April 2020, a new clinical entity was observed in Europe [2,3] with some similarities to severe forms of Kawasaki disease (KD), Kawasaki disease shock syndrome (KDSS) and toxic shock syndrome (TSS), but also with some peculiarities [3].

This novel syndrome was defined as “multisystem inflammatory syndrome in children (MIS-C)” by the US Centers for Disease Control and Prevention and World Health Organization [4,5], and “paediatric multisystem inflammatory syndrome temporally associated with COVID-19 (PIMS)” by the Royal College of Pediatrics and Child Health [6]. Patients with MIS-C present with persistent fever, inflammatory markers elevation and multi-organ dysfunction due to systemic inflammation. A temporal correlation with a recent SARS-CoV-2 infection (history of close contact and/or positive serology) characterize this syndrome. The cardiological involvement is predominant in MIS-C and ranges from mild to severe left ventricular dysfunction, arrythmias, coronary artery dilatations or aneurysms and cardiogenic shock [7,8].

Here, we describe the cardiovascular and clinical manifestations, laboratory findings and cardiac imaging of MIS-C observed in our center over a one-year period. The aim of the study is to point out the distinct cardiovascular involvement among different age groups. 

## 2. Materials and Methods

We retrospectively analyzed patients with a diagnosis of MIS-C, according to the CDC case definition [4], admitted to the Gaslini Institute of Genoa between March 2020 and September 2021. Our children’s hospital is a referral center for cases of MIS-C in the Liguria region (1,500,000 inhabitants). Data were obtained from Galileo software (NoemaLife We care) and included in the international registry on COVID-19-related hyperinflammation in children and young adults (HyperPED-COVID), after informed consent. The study was approved by the Ethical Review Board of Regione Liguria. 

All patients less than 21 years of age with a positive test for SARS-CoV-2 (serology (IgG title) or reverse transcriptase-polymerase chain reaction (RT-PCR) from a nasopharyngeal swab) or close contact with COVID-19 positive cases were included. Our sample was stratified into two groups, according to age: younger (group 1) and older than 6 years old (group 2). For each patient we collected demographic data, past medical history and clinical manifestations, laboratory findings and diagnostic procedures. Lymphocytes were expressed as a total count and as a ratio in respect to the lower normal values for age (2600/mmc for children < 6 years, 2000 for children ≥ 6 years). Cardiologic investigations and admission included: cardiac enzymes with the N-terminal prohormone of brain natriuretic peptide (NT-pro-BNP), a 12-lead electrocardiogram (ECG) and an echocardiogram (echo). In case of cardiologic involvement, patients were monitored with cardiac telemetry and daily clinical evaluation. Cardiac evaluations (ECG and echo) were generally performed daily in the acute phase and, then, according to the clinical status until clinical improvement. Cardiac magnetic resonance imaging (cMRI) was reserved for selected cases with moderate to severe left ventricular dysfunction. 

The transthoracic echocardiogram was performed using an EpiqCVx machine (Philips Medical Systems, Milan, Italy). Left ventricular ejection fraction (LVEF) was evaluated by using Simpson’s biplane method. Left ventricular (LV) dysfunction was defined as mild with LVEF between 41–52%, moderate with LVEF between 30–40% and severe if <30% [9]. Coronary artery involvement was established by internal measurement based on the Boston z-score system. We classified coronary artery anomalies based on the American Heart Association z-score classification: dilatation, “z-score 2–2.5”; small aneurysm, “z-score 2.5–5”; medium aneurysm, “z-score 5–10”; and giant aneurysm, “z-score > 10” or internal measurement > 8 mm [10]. We also analyzed the presence of mitral valve regurgitation and the presence of pericardial effusion. The cMRI was performed on an MRI scanner Achieva 1.5 Tesla (Philips Medical System, Best, The Netherlands). The study protocol included the assessment of biventricular EF% and regional kinesis, evaluation for myocardial edema and hyperemia, and the study of the pericardium.

Statistical analysis was performed using Microsoft Excel (Microsoft Corporation, Milan, Italy, EU) and software R (R Foundation for Statistical Computing, Vienna, Austria). Continuous variables were summarized as medians and IQRs, and categorical variables were presented as frequencies and percentages. A chi-square test and Fisher’s exact test were used for categorical variables, whereas a Mann–Whitney U test was performed for continuous variables. We considered a statistically significant *p* value to be < 0.05.

## 3. Results

This study included 25 patients who met CDC criteria for MIS-C, 13 (52%) were younger than 6 years of age and 12 (48%) were older. All patients were previously healthy. The median age at the onset of illness was 5 years (IQR 3, 12) and median time from fever onset to hospital admission was 3 days (IQR 3, 4). Diagnosis of MIS-C was performed within a median of 2 days (IQR 1, 3). We recorded a homogeneous prevalence between male and female (56% vs. 44%, respectively). The cohort was predominantly European Caucasian (60%), then Hispanic (20%), African (12%) and Asiatic (8%). Most patients had a positive COVID-19 test: serology, No. 20 (80%); PCR nasopharyngeal swab, No. 3 (12%), whereas 2 patients (8%) had close contact with a COVID-19 positive case. The prevalence of positive serology was higher in group 2 than group 1. The demographic data, COVID-19 tests, clinical manifestations and therapy are summarized in Table 1.

A severe increase of inflammatory markers was present in both groups with statistically significant evidence of hyperferritinemia (*p <* 0.012) in group 2 (Figure 1a). Lymphopenia is observed in both groups, showing a ratio between the total lymphocytes and the lower normal limit for age that was statistically significantly lower in group 2 (*p* < 0.014) (Figure 1b). All patients had NT-pro-BNP markedly elevated, with no difference between the two groups, whereas Troponin I elevation was mild and more pronounced in group 2 than in group 1 (*p* < 0.042) (Table 2).

All patients displayed some cardiovascular manifestations, as shown in Table 3, with some differences between the two groups (Table 4). Hypotension at admission was observed in seven patients and was more frequent in group 2 (*p <* 0.022): four cases displayed a hypovolemic shock. No cardiogenic shock was observed.

Pathological ECG was found in 60% of cases. The most common anomalies observed were the ST segment-T wave (56%) and prolonged QTc segment (40%), which were more frequent in group 2 (*p <* 0.008 and *p <* 0.018, respectively). The median time from disease onset and the identification of a prolonged QTc was 8.5 days (IQR 8, 10), with a maximum value of 530msec. Continuous ECG monitoring did not record significant arrhythmias. 

All patients presented an abnormal echocardiogram performed within a median of 5 days (IQR 4, 6) from the disease onset. The median LVEF was 55% (IQR 50, 60) but in 40% of cases LV dysfunction (LVEF < 53%) was documented. Coronary involvement was found in 52% of patients, including dilatation and/or aneurysms, and pericardial effusion was found in almost all. We detected a peculiar distribution of the cardiac involvement according to age. In patients < 6 years, there prevailed a coronary involvement (*p <* 0.024), either dilatation (30.7%) or a small aneurysm (30.7%). In only one case a medium aneurysm was detected. Most of them showed preserved myocardial function. Conversely, in patients ≥ 6 years we observed a prevalence of LV dysfunction (66.7%), generally from mild to moderate. In two cases only were coronary dilatations observed. 

Patients with myocardial impairment underwent cMRI at a median time of 13 days (IQR 12, 17) from the symptoms’ onset. We documented a median LVEF of 60% (IQR 60, 63), myocardial edema in two patients and pericardial effusion in all patients.

Data analysis showed that patients with a higher fibrinogen level had worse LVEF than others (Figure 2a). We also observed that the NT-pro-BNP value appeared inversely related to LVEF (Figure 2b). Troponin I did not correlate with the severity of cardiac involvement.

The management of MIS-C patients was established according to the severity of the clinical picture. The treatment was administered within the first 48 h from the admission, as soon as the diagnosis was confirmed, with a median time between fever onset and hospital admission of 3 days (IQR 3, 4). All patients received high doses of intravenous immunoglobulin (IVIG), plus IV steroids (Table 1). Anakinra (IL-1 receptor antagonist) was given to eight patients (one in group 1, seven in group 2) with a severe condition and/or refractory to the initial treatment. In most cases, low dose aspirin was administered, and associated to anticoagulation with heparin in 68% of cases. Heparin was given at a prophylactic or therapeutic dosage, according to individual thrombotic risk. Sixteen patients were treated with cardioactive drugs (beta-blockers, ACE-inhibitors, diuretics). No one needed intensive care unit admission, inotropes and/or invasive mechanical ventilation. 

Median hospitalization length was 20 days (range 14.5, 26.5). ECG alterations were completely resolved in all cases, except in one with a persistent, prolonged QTc segment. Moreover, an echo examination at discharge showed a complete recovery of LVEF and regression of coronary dilatations and small aneurysms; in one patient a medium aneurysm persisted after hospitalization.

## 4. Discussion

MIS-C is now a well-defined pediatric disease, developed during the COVID-19 pandemic, and characterized by a hyperinflammatory condition with severe multisystem involvement associated with SARS-CoV-2 [11]. Different global health organizations have established diagnostic criteria that include: persistent fever (T > 38 °C), systemic inflammatory state with elevation of inflammation markers, neutrophilic leukocytosis, lymphopenia and organ dysfunction with laboratory or epidemiological evidence of a past SARS-CoV-2 infection after the exclusion of other microbiological causes [4,5,6].

Similarities between MIS-C and KD have been found, such as persistent fever and high prevalence of oral mucositis, conjunctivitis and skin rash. However, gastrointestinal symptoms and cardiac dysfunction with shock are significantly more common in patients with MIS-C than in those with KD [12]. 

In our cohort, males and females were homogenously hit by the illness. The median age of disease onset was 5 years, lower than the median age of 8–9 years reported by other European studies [13,14,15,16]. As our sample included a wide spectrum of age, we decided to analyze the cardiovascular manifestations according to age. Thus, we divided our population into two groups: preschoolage (<6 years) and schoolage (≥6 years).

It was highlighted that there was a constant increase of the inflammatory markers D-dimer and fibrinogen, in particular in group 2. We found that a higher fibrinogen level was related to the worst LVEF. We suggest that fibrinogen could be the expression of a hyperinflammatory state, responsible for myocardial damage that is phlogosis-induced. In most of the patients, the number of total platelets was lower than expected in classical KD. Lymphopenia represented MIS-C’s most common hematological abnormality and was observed in both groups, with a negative ratio in respect to lower normal levels for age. As it is reported in the literature data, troponin elevation is frequent in cases of myocardial injury [17,18,19]. Indeed, in our cohort troponin I was higher in school-age patients who presented more frequent LV dysfunction. The NT-pro-BNP was markedly elevated even after hospital admission, especially in patients with impaired LVEF. The data could relate to LV dysfunction severity, as reported by Kavurt et al. [18], who described higher levels of NT-pro-BNP in patients with biventricular dysfunction than in those with preserved myocardial function.

Cardiac involvement represents one of the most frequent manifestations in MIS-C, found in up to 84% of cases [18]. In our study, all patients presented cardiovascular complications, with a different distribution according to age. Hypotension at admission was more frequent in older patients than in younger ones, and only four cases evolved into hypovolemic shock, responsive to fluid replacement. No cases of cardiogenic shock were reported, and nobody needed inotropic support. These data differ from the literature, in which a prevalence of vasoplegic and/or cardiogenic shock were observed in 40% to 80% of MIS-C cases [8,20]. We hypothesize that this favorable result was mainly due to the early and multi-step therapeutic approach used in our center [21] that was modulated on the disease severity. 

MIS-C related ECG changes are well described, including repolarization abnormalities, sinus bradycardia, AV block and, rarely, tachyarrhythmias [22]. We recorded typical ECG alteration and three cases of first-degree atrioventricular block that resolved during hospitalization. Prolonged QTc is usually reported as being due to electrolyte abnormalities or specific drugs intake [23]; only Dionne et al. [22] described spontaneous and transient QTc prolongation as we observed in our cohort. QTc prolongation has also been reported in acute myocarditis and seems to identify the most severe cases associated with the worst prognosis [24,25,26,27]. We can speculate that a diffuse myocardial involvement and inflammation in MIS-C could contribute to (or be responsible for) QTc prolongation. Nonetheless, the early and aggressive anti-inflammatory treatment could have had a protective effect against both arrhythmic events and myocardial function deterioration.

Echocardiograms showed pathological findings at the initial assessment in most of the patients. Mild to moderate pericardial effusion and atrioventricular valve regurgitation were frequently documented, whereas myocardial impairment was found in 40% of cases. On the other hand, the incidence of coronary involvement was about 50%, much higher than described in other MIS-C case series (about 21.7%) [28]. This datum seems to be more similar to KD, in which coronary involvement is reported between 23–50% [29].

Data analysis, focused on comparison between different age groups, revealed interesting findings. 

Preschool patients presented clearly prevalent coronary involvement compared to those of schoolage (*p* < 0.024), characterized by dilatation, small aneurysms and, in only one case, a medium aneurysm. This evidence agreed with that of Rakha et al. [30], that found predominant coronary changes (about 50% of cases) in his cohort, made up of only MIS-C patients younger than 5 years. Other frequent clinical signs in this subgroup were cheilitis, conjunctivitis, alterations of the extremities and cervical lymphadenopathy, mimicking a pattern of presentation of a Kawasaki-like disease. However, among the 13 patients < 6 years of age, 6 did not fulfill the KD criteria. All 7 patients that satisfied the KD criteria were positive for serology (5 patients) or had a clear history of familiar contact (2 patients). This issue, in combination with the considerable epidemiological context we had in Italy during the pandemic, is the stronger argument supporting the diagnosis of MIS-C in our patients. Moreover, they also presented a higher frequency of gastrointestinal involvement, an overall reduction of circulating lymphocytes in respect to normal levels for their age and a lack in the increase of platelets, which are not typical findings in “classical KD” [31,32].

The pattern of coronary changes, marked by dilatation, would suggest that it may be the result of a response to the underlying hyperinflammatory condition, which occurs in MIS-C, rather than the typical alterations that occur in KD, culminating with the destruction of vascular walls [14,33,34,35]. This hypothesis is further supported by the evidence of gradual regression of coronary dilatation with progressive clinical and biohumoral improvement until complete recovery at the time of discharge.

Differently, in the school-age group, myocardial involvement dominates, characterized by a lower median LVEF with mild to moderate dysfunction in more than half of patients. These data agree with those reported in the literature [28]. The prevalent myocardial involvement in older patients appears to be more consistent with the clinical manifestations typical of MIS-C.

We suggest that the prevalent Kawasaki-like presentation in the younger subgroup may result from a different, age-linked immune response to a virus in a genetically susceptible host. Although the exact etiopathogenesis of MIS-C and KD remains unclear, it is hypothesized that it is an immune-mediated process triggered from a virus infection in both conditions [36,37,38]. Even if the two entities present some overlapping clinical and immunological features, they develop a distinct inflammatory reaction [39]. Maybe the immune response to SARS-CoV-2 is different in younger children in relation to a cross-reactive immunity to other coronaviruses [40]. Moreover, it could be possible that the inflammatory reaction is less violent in preschoolages, according to the age-related immune system immaturity [41]. In comparison, older patients with “typical” MIS-C presentation develop a severe hyperinflammatory reaction, possibly due to silent inborn errors of immunity triggered by SARS-CoV-2, as described by Sancho-Shimizu et al. [36]. The resulting uncontrolled inflammatory response could be the main reason for cardiomyocyte injury, often observed in this age group. Another possibility is the cells could have a distinct tropism for the virus based on age. Younger children could have more cell markers inviting the virus into the coronaries than into the myocardium, and older children could have the opposite. Indeed, it is described that SARS-CoV-2 receptors are widely expressed in organs with a possibly different distribution, maturity and function in relation to children’s age [42,43,44]. Further studies are needed to better explain our hypothesis.

The cMRI was performed on selected patients with moderate to severe cardiovascular impairment after about 2 weeks from disease onset. At the examination, we documented pericardial involvement and normal biventricular function in almost all cases. Myocardial edema was identified in only two cases. Webster et al. [45] demonstrated normal cardiac findings at a follow-up cMRI after 2 months. These data suggest a possible immuno-mediated origin of myocardial damage during the acute phase, with rapid improvement thanks to targeted immunomodulatory therapy.

Based on the severity of clinical manifestations shown by the patients, we used a prompt multi-step anti-inflammatory treatment protocol, as described by Brisca et al. [21]. By using this approach, none of the patients treated required intensive care unit admission and no deaths occurred. Moreover, the combination of IVIG and intravenous steroids seems to determine a lower risk of cardiovascular dysfunction compared to IVIG alone, as we demonstrated [46]. Our satisfactory results differ from most of the literature data, which describe higher prevalence of ICU admission (60–80% of cases), hemodynamic support (12–47% of cases) and mechanical ventilation (5–49%of cases) [15,47,48,49]. The rationale underlying this protocol was to use an early aggressive treatment in MIS-C patients with modulated therapeutic interventions based on the clinical severity. Using this therapeutic strategy, we helped to prevent the progression of the inflammatory process and cardiac impairment, and to avoid admission to the ICU.

Our cohort had a favorable early outcome with normalization of cardiac enzymes, improvement of LVEF about 1 week after treatment initiation and complete recovery at discharge. Some case series report normal LVEF at discharge, even in the presence of reduced values of global longitudinal strain (GLS), which maybe suggestive of a persistent subclinical myocardial dysfunction [2,50]. The natural history of coronary involvement is uncertain, but we documented a full resolution of coronary changes, except in one case of persistent medium coronary aneurysm.

## 5. Conclusions

Cardiovascular manifestations are very common in MIS-C and are presented with a peculiar distribution according to different age groups: predominant coronary involvement in children younger than 6 years old and myopericardial involvement in older ones. Prompt and aggressive anti-inflammatory, multitarget therapy has contributed to a favorable outcome, preventing the evolution to multiorgan failure and death.

## 6. Study Limitations

Our study presents some limitations, which include the retrospective nature of the study and the small sample size. An additional limitation represented by the lack of GLS analysis that could have helped the clinicians to detect subclinical cardiac damage for better treatment and follow-up. Further studies are needed to examine the long-term outcome for MIS-C patients.

## Figures and Tables

**Figure 1 children-09-00583-f001:**
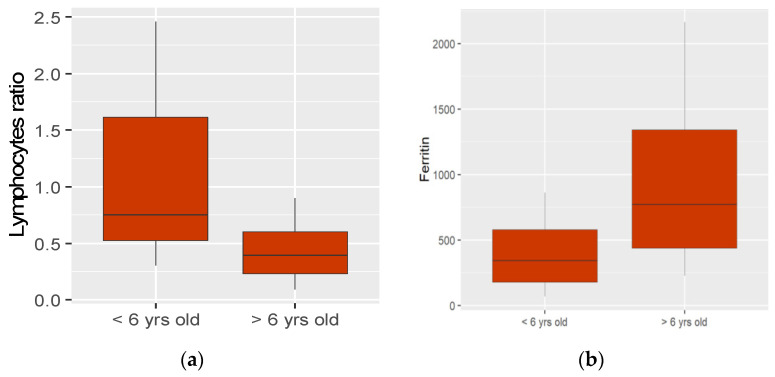
Laboratory findings: lymphocytes ratio and ferritin. (**a**) Lymphocytes ratio between the total lymphocytes and the lower normal limit for age was statistically significantly lower in group 2 (≥6 years) than in group 1 (<6 years); (**b**) Ferritin statistically significantly higher in group 2 (≥6 years) than in group 1 (<6 years). Yrs old: years old.

**Figure 2 children-09-00583-f002:**
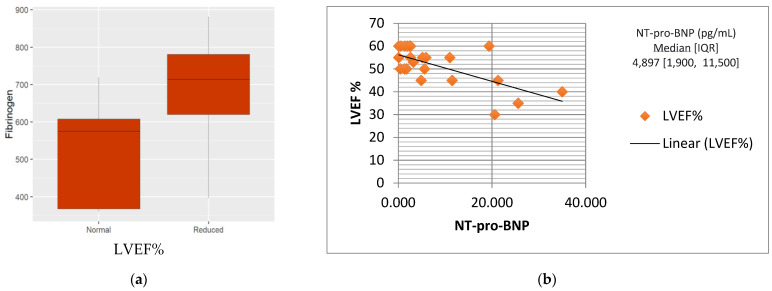
(**a**) Fibrinogen levels significantly higher in reduced LVEF group. (**b**) Trend of LVEF, according to NT-pro-BNP levels. LVEF: left ventricular ejection fraction; IQR: interquartile range; NT-pro-BNP: N-terminal prohormone of brain natriuretic peptide.

**Table 1 children-09-00583-t001:** Demographics, clinical features, COVID-19 tests and therapy of MIS-C patients, according to different age groups. Group 1: <6 years old (No. 13); group 2: ≥6 years old (No. 12). MIS-C: multisystem inflammatory syndrome in children.

MIS-C Cohort	Total Patients 25 (100%)	Group 1 <6 Years 13 (52%)	Group 2 ≥6 Years 12 (48%)
Female	11 (44)	6 (46.2)	5 (41.7)
Male	14 (56)	7 (53.8)	7 (58.3)
Ethnicity			
Caucasian	16 (64)	7 (53.8)	9 (75)
Hispanic	5 (20)	4 (30.8)	1 (8.3)
African	2 (8)	2 (15.4)	0 (0)
Asian	2 (8)	0 (0)	2 (16.7)
COVID-19 tests			
Positive SARS-CoV-2 nasopharyngeal swab	3 (12)	2 (15.4)	1 (8.3)
Positive SARS-CoV-2 IgG title	20 (80)	9 (69.2)	11 (91.7)
Contact with patients with COVID-19	2 (8)	2 (15.4)	0 (0)
Clinical manifestations			
Cardiological	25 (100)	13 (100)	12 (100)
Respiratory	9 (36)	2 (15.4)	7 (58.3)
Gastrointestinal	22 (88)	12 (92.3)	10 (83.3)
Muco-cutaneous	22 (88)	12 (92.3)	10 (83.3)
Adenopathy	14 (56)	7 (53.8)	7 (58.3)
Therapy			
IVIG	25 (100)	13 (100)	12 (100)
Steroid	24 (96)	12 (92.3)	12 (100)
Anakinra	8 (32)	1 (7.7)	7 (58.3)
Aspirin	23 (92)	13 (100)	10 (83.3)
Low molecular weight heparin	17 (68)	6 (46.1)	11 (91.7)
Beta-blockers	6 (24)	2 (15.4)	4 (33.3)
Diuretics/anti-congestive heart failure drugs	16 (64)	7 (53.8)	9 (75)
Inotropic drugs	0 (0)	0 (0)	0 (0)
Antibiotics	14 (56)	5 (38.5)	9 (75)

SARS-CoV-2: severe acute respiratory syndrome coronavirus 2. Cardiological (hypotension, ECG alterations, echo alterations); respiratory (dyspnea, chest pain, cough, pleuritis, pleural effusion); gastrointestinal (vomit, abdominal pain, diarrhea, ascites); muco-cutaneous (cheilitis, conjunctivitis, skin rashes, erythematous pharyngitis, glossitis); adenopathy (generalized or cervical lymphnodes enlargement, other lymphadenopathy). IQR: interquartile range. IVIG: intravenous immunoglobulin.

**Table 2 children-09-00583-t002:** Laboratory findings in the MIS-C cohort according to different age groups. Group 1: <6 years old (No. 13); group 2: ≥6 years old (No. 12). Reported only significant *p* values (<0.05).

Laboratory Findings	Group 1 (<6 Years) Median [IQR]	Group 2 (≥6 Years) Median [IQR]	*p* Value
ESR (1–10 mm/h)	47 [29, 66]	66 [55, 78]	
CRP (<0.46 mg/dL)	12.6 [8.0, 19.3]	20.3 [12.3, 25.2]	
PCT (<0.5 ng/mL)	5.4 [4.2, 7.8]	4.2 [2.5, 13.3]	
WBC (4–9.8 cell/mm^3^)	19.4 [17.3, 21.5]	15.6 [12.9, 19.8]	
Neutrofils (2–6.4 cell/mm^3^)	11.9 [9.8, 16.7]	14.1 [11.9, 16.1]	
Lymphocytes (2.0–5.8 cell/mm^3^)	1.6 [1.0, 2.8] r 0.5	0.6 [0.3, 0.9] r 0.275	<0.002 <0.014
Hb (11.5–16.5 g/dL)	12.2 [11.1, 12.6]	13.1 [12.5, 14]	<0.024
Platelets (150–450 cell/mm^3^)	167.000 [103.000, 361.000]	136.500 [109.750, 182.000]	
Fibrinogen (180–350 mg/dL)	564.0 [373.0, 662.0]	655.0 [533.8, 840.8]	
D Dimer (<0.55 mcg/dL)	3.0 [1.6, 4.1]	3.8 [2.3, 6.7]	
Albumin (3.800–5.400 mg/dL)	3000 [2.500, 3.300]	3100 [2.600, 3.400]	
Ferritin (20–200 ng/mL)	343.0 [177, 578]	769.5 [437.5, 1340]	<0.012
Troponin I (<0.16 ng/mL)	0.1 [0.1, 0.1]	0.2 [0.1, 0.4]	<0.042
CK MB (<4.9 UI/L)	2.4 [1.6, 3.6]	3.0 [1.0, 4.2]	
NT-pro-BNP (<150 pg/mL)	3899.5 [2135.2, 9289.5]	6329.5 [2444.2, 14,848]	

IQR: interquartile range; y: years; ESR: erythrocyte sedimentation rate; CRP: C-reactive protein; PCT: procalcitonin; WBC: white blood cells; r: ratio; Hb: hemoglobin; CK-MB: creatine-kinase MB; NT-pro-BNP: N-Terminal prohormone of brain natriuretic peptide.

**Table 3 children-09-00583-t003:** Cardiovascular manifestations in the MIS-C cohort. MIS-C: multisystem inflammatory syndrome in children.

Cardiovascular Manifestations		No.(%)
Electrocardiogram abnormalities		15 (60)
Abnormal ST-T wave segment		14 (56)
Prolonged QTc interval		10 (40)
Bradicardia		3 (12)
Atrioventricular Block		2 (8)
Non-sustained supraventricular/ventricular tachyarrhythmias		3 (12)
Echocardiogram anomalies		25 (100)
Coronary involvement		13 (52)
	Dilatation (z-score + 2 − 2.5)	7 (28)
	Small aneurysm (z-score + 2.5 − 5)	4 (16)
	Medium aneurysm (z-score + 5 − 10)	1 (4)
LVEF% median [IQR]		55 [50, 60]
LVEF > 53%		15 (60)
LVEF < 53%		10 (40)
	EF 41–52%	7 (28)
	EF 30–40%	3 (12)
	EF < 30%	0 (0)
Pericardial effusion		20 (80)
Mitral regurgitation		10 (40)

IQR: interquartile range. LVEF: left ventricular ejection fraction. EF: ejection fraction.

**Table 4 children-09-00583-t004:** Detailed cardiovascular manifestations comparing different age groups. Group 1: <6 years old (No.13); group 2: ≥6 years old (No.12). Reported only significant *p* values (<0.05). MIS-C: multisystem inflammatory syndrome in children.

Cardiovascular Manifestations	Group 1 (No.%)	Group 2 (No.%)	*p* Value
Hypotension	1 (7.7)	7 (58.3)	<0.022
Electrocardiogram abnormalities	4 (30.7)	11 (91.6)	<0.006
Abnormal ST-T wave segment	4 (30.7)	10 (98.3)	<0.008
Prolonged QTc interval	2 (15.4)	8 (66.6)	<0.018
Bradicardia	1 (7.6)	2 (16.6)	
Atrioventricular Block	1 (7.6)	1 (8.3)	
Non-sustained supraventricular/ventricular tachyarrhythmias	0 (0)	3 (25)	<0.05
Echocardiogram anomalies	13 (100)	12 (100)	
Coronaric involvement	10 (77)	2 (16.6)	<0.024
Dilatation (z-score + 2 − 2.5) Small aneurysm (z-score + 2.5 − 5) Medium aneurysm (z-score + 5 − 10)	4 (30.7)	2 (100)	
	4 (30.7)	0 (0)	
	1 (7.6)	0 (0)	
LVEF% median [IQR]	55 [50, 60]	51 [45, 60]	
LVEF >53%	11 (84.6)	4 (33.3)	
LVEF <53%	2 (15.4)	8 (66.7)	
EF 52–41% EF 40–30% EF <30%	2 (15.4)	5 (41.7)	
	0 (0)	3 (25)	
	0 (0)	0 (0)	
Pericardial effusion	8 (61.5)	12 (100)	
Mitral regurgitation	3 (23.1)	7 (58.3)	

IQR: interquartile range. LVEF: left ventricular ejection fraction. EF: ejection fraction.

## Data Availability

The data presented in this study are available on request from the corresponding author. The data are not publicly available due to Institutional and Research policies.

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
