# Peer review of "Cardiovascular Manifestations in Multisystem Inflammatory Syndrome in Children (MIS-C) Associated with COVID-19 According to Age"

_children, 2022, doi:10.3390/children9050583_

Round 1

Reviewer 1 Report

The authors Campanello et al have written a descriptive study on the cardiovascular manifestations noted in MIS-C associated with COVID-19. Interestingly, children younger than 6 years of age were noted to have mroe coronary abnormalities, while older ones were noted to have predominantly myopericardial involvement. This study has merit, however there are several grammatical and spelling errors.  There are a few major comments I have mentioned as well. 

Major comments:

  1. How did you classify the type of shock? Were PA catheters placed?
  2. In the discussion, instead of simply stating your results, I would recommend speculating on the rationale and pathophysiology for your findings. 

Minor comments: There are SEVERAL grammatical errors, errors in tenses, spellings, spacing. Here are a few:

Line 38 - Capitalize Severe Acute Respiratory Syndrome Coronavirus

Line 48 - present "with" persistent fever 

Line 54 - over "a" one year period

Line 62 - capitalize Ethical Review Board 

Line 63 - Please rephrase the sentence to "All patients less than 21 years of age ...positive cases were included"

Line 67 - correct the spelling of demographic 

Line 69 - Keep the tense the same - "included"

Line 69 - Please mention the full form of NT-proBNP once before abbreviating it 

Line 74 - no need to capitalize magnetic resonance imaging 

Line 75 - reserved for 

Line 70 - no need to capitalize echo 

Line 81 - correct the tense - "established" 

Lines 82 and 83 - rephrase sentence to "we classified coronary artery anomalies based on the American Heart Association z-score classification"

Line  82 - the spelling of heart is incorrect 

Line 85 - rephrase "we've also analyzed" to " we also analyzed" 

Line 95, 98 - please correct tense 

Line 98 - no need to write "n"

Line 102 - Correct grammar 

Line 104 - should be "then", not "than"

Author Response

Response to Reviewer 1

The authors Campanello et al have written a descriptive study on the cardiovascular manifestations noted in MIS-C associated with COVID-19. Interestingly, children younger than 6 years of age were noted to have more coronary abnormalities, while older ones were noted to have predominantly myopericardial involvement. This study has merit, however there are several grammatical and spelling errors.  There are a few major comments I have mentioned as well.

We have revised the manuscript, according your suggestion related to grammatical and spelling errors.

Major comments:

  1. How did you classify the type of shock? Were PA catheters placed?

Response 1: We thank the reviewer for this interesting comment. We classify the type of shock on the base ofsymptoms, physical exam, vital signs and imaging (echocardiogram and abdominal ultrasound).

We placed PA catheters in patients with poor clinical condition and hypotension at admission. While, we used an automatic blood pressure monitor - with appropriate cuff bladder size for age – for patients with normal blood pressure at admission.

We had 4 case of hypovolemic shock, characterized by:

-fever,poor liquid intake and vomit/diarrhea;

- systolic pressure at least fifth percentile for age: 70mmHg +[2x age in years] in children 1 month to 10 years or <90 mmHg in children 10 years of age or older;

-alteration of skin perfusion (capillary refill>2”);

-reduced urine output (0.5-1 ml/kg per hour);

- hypoalbuminemia (cause of extracellular fluid loss);

-no heart failure.

This patientshad a good response to intravenous fluid resuscitation (crystalloid or colloid solutions).

No case of cardiogenic shockwas found, by means of severe impairment of myocardial performance with end‐organ hypoperfusion and hypoxia. Moreover, no evidence of hypotension refractory to volume resuscitation requiring pharmacological or mechanical intervention.

  1. In the discussion, instead of simply stating your results, I would recommend speculating on the rationale and pathophysiology for your findings. 

Response 2: The discussion has been revised according to your suggestion.

Please see the attachment for the revised manuscript.

Reviewer 2 Report

The authors present a single-center retrospective observational study of 25 children with MIS-C. They described their population and focused on the cardiovascular manifestations in the subgroup < 6 years old compared to the subgroup > 6 years old. The main clinical message is that there is a significant difference in cardiac presentation of MIS-C patients int the preschoolers age versus the older with the first presenting with coronary arteries involvements and the latter with cardiac dysfunction. 

First of all, the manuscript needs some editing, principally related to grammar, syntax, typos and diction, some sentences are farraginous and unclear. See some comments below.

The main finding of your study is the different cardiological presentation between the two subgroups?  How can you explain the difference? Do you have any hypothesis? Please add some comments in the discussion. My major concern related to your results are the followings. You described a population that is significant younger than those reported in literature with a median age of 5 years old compared to a median of 8 years old in the other papers. Moreover, in the younger subgroup you found data that are very similar to those reported in Kawasaki disease. Could you considered that maybe some of your young children can be diagnosed more with Kawasaki than with MIS-C? Could you prove or motivated better how did you reach the differential diagnosis between the two diseases? Maybe you can add IL-1 vs IL-6 or you can add some information’s about the clinical presentation (for example abdominal involvement is highly related to MIS-C and rarely there are all criteria for Kawasaki) and the number of uncertain diagnosis. Could you please report the positive serology (“only 76%” ) divided into the two subgroups? Serology is a significant marker of a recent COVD-19 infection and can help to discriminate among diseases. It would be nice to know the prevalence of IgG/IgM positivity in the subgroups. Please add some clinical information divided into the two subgroups, to help the reader to discriminate better.

Some additional comments and a list of some typos or farraginous sentences ( the list is not exhaustive).

Introduction

Line 37: inflammatory(space)syndrome

Line 49: related to acuteà MIS-C is related to previous COVD contact, not to the acute illness (even if some patients can still have a residual swab positivity)

Line 54: THE metropolitan area

Line 58: for case of MIS-C IN Liguria (Italian region)

Line 70-74: rewrite the sentence à too long and incorrect. Subjected to telemetry (ex. patients were monitored with…after daily clinical evaluation (POINT). Cardiac evaluations were generally performed daily in the acute phase and then accordingly to the clinical status)

Line 77: by THE pediatric cardiologist (not capital letter) by using AN EpiqCvx machine

Line 80: remove % after LVEF. The sentence could be easier if written like that: LV dysfunction was defined as mild with LVEF between 41-52%, moderate with LVEF between 30-40% (invert!) and severe if < 30%.

Line 81: measurement

Line 83: we have considered

Line 83-84 : please put the z score in brackets

Line 85: We’ve also analyzed

Line 98-99: remove n.

Line 100: THE median age at the onset OF ILLNESS was

Line 102: a little difference between male and female à It does not make sense. Rewrite (ex. We recorded a homogeneous prevalence between male and female OR Males and females were homogenously hit by the illness.

In Table 1 and in the text: you wrote IGIV à please correct in IVIG (IntraVenous ImmunoGlobulin)

Figure 1 : in the comment à higHer (not higer).

Line 137: in the majority of casewe observed A PREVALENCE OF

Line 142: higHer (please correct in all the text)

Line 143: inversely RELATED to

Line 164-165 please specify if prophylaxis or therapeutic dosage

Line 186-187: In our cohort there was no evidence of difference between male and female (????). What do you mean? Did you compare cardiac results between male and female? Or do you what to say that there is not a sex prevalence in the incidence of illness? Please rewrite

Line 196-201: too farraginous: REWRITE.

Line 209: neither is incorrect. You can write…fluid replacement. No case of cardiogenic shock were reported and nobody needed inotropic support.

Line 217: in the literature (delate, redundant)

Line 225: against (space) arrhythmic

Line 230-233: please add a possible explanation and/or how you can or cannot differentiate the two disease.

Line 235: interesting(space) findings

Line 261: we have used

Line 264: while à AND in THE noT-responder

Line 271: accordingly.

Line 272: from THE admission

Line 275-276: with MODULATED therapeutic interventions modulated based

Line 281-283: what do you mean? Please, rewrite.

Author Response

Response to Reviewer 2

The authors present a single-center retrospective observational study of 25 children with MIS-C. They described their population and focused on the cardiovascular manifestations in the subgroup < 6 years old compared to the subgroup > 6 years old. The main clinical message is that there is a significant difference in cardiac presentation of MIS-C patientsin the preschoolers age versus the older with the first presenting with coronary arteries involvements and the latter with cardiac dysfunction.

First of all, the manuscript needs some editing, principally related to grammar, syntax, typos and diction, some sentences are farraginous and unclear.

We have revised the manuscript, according your suggestion related to grammar, syntax, diction and sentences.

Major comments:

  1. The main finding of your study is the different cardiological presentation between the two subgroups? 

Response 1: Yes, it is. We tried to better clarify this point in theintroduction (line 20-23).

  1. How can you explain the difference? Do you have any hypothesis? Please add some comments in the discussion.

Response 2: A possible interpretation and explanation of our findings are now present in the discussion (line233-252).

My major concern related to your results are the followings. You described a population that is significant younger than those reported in literature with a median age of 5 years old compared to a median of 8 years old in the other papers. Moreover, in the younger subgroup you found data that are very similar to those reported in Kawasaki disease.

  1. Could you considered that maybe some of your young children can be diagnosed more with Kawasaki than with MIS-C?
  2. Could you prove or motivated better how did you reach the differential diagnosis between the two diseases? Maybe you can add IL-1 vs IL-6 or you can add some information’s about the clinical presentation (for example abdominal involvement is highly related to MIS-C and rarely there are all criteria for Kawasaki) and the number of uncertain diagnosis.

Reply to point 3 and 4:

We thank the reviewer for this interesting comment. Indeed our study confirms that younger children with MIS-C display a clinical picture very similar to KD. However, among the 13  patients<6years of age, 6 didn’t fulfill the KD criteria. All the 7patients satisfied the KD criteria were positive for serology (5 pts) or had a clear history of familiar contact (2 pts). This issue, in combination with the strong epidemiological context we had in Italy during the pandemic, is the stronger argument supporting the diagnosis of MIS-C in our patients. Moreover, they also presented anhigher frequency of gastrointestinal involvement,  an overall reduction of circulating lymphocytes in respect to normal levels for age and a lack in the increase of platelets, that are not typical findings in “classical KD”. In the new version of the resultsthe ratio between the total lymphocytes and the lower normal limit for age is now shown, showing the clear presence of a lymphopeniain younger children also.In the discussion, we point out the overlap between KD and MIS-C in younger children, pointing out the most relevant similarities and differences (Discussion line 8-11, line 21-24, line 67-82 ).

Despite the fact that the study is mainly focused to described the most relevant cardiovascular manifestations of  MIS-C, we recognize that this point is of interest. We tried to clarify better this issue both in the results and in the discussion.

We have a long experience in measuring cytokine production in inflammatory conditions. IL-6 is usually elevated during any systemic inflammation in a nonspecific manner. Conversely, IL-1 is quite difficult to detect in sera, even in classical IL-1 mediated disease, such as NLRP3-mediated auto inflammatory diseases. The proper method of evaluation of the possible pathogenic relevance of IL-1 should be based on the study of the secretion of IL-1 by whole blood or monocytes upon specific stimulation with LPS or other stimuli. Based on these considerations and previous experiences,  we choose not to test serum cytokines in this context.

  1. Could you please report the positive serology (“only 76%” ) divided into the two subgroups? Serology is a significant marker of a recent COVID-19 infection and can help to discriminate among diseases. Please, add some clinical information divided into the two subgroups, to help the reader to discriminate better.

Response 5:

The distribution of a positive serology was the following:

  • Group 1 (<6y): positive serology in 9; 2 positive RT-PCR; 2 case of familiar contact;

Group 2 (³6y): positive serology in 11; 1 positive RT-PCR.

Indeed the prevalence of positive serology was higher in group 2 than group 1 . We want to point out that in 2 patients of group 1 the serological test was not available because they display their disease onset at the beginning of pandemic, when the serological tests were not yet available or reliable. However, both patients presented negative RT-PCR at nasopharyngeal swab but a positive close contact. In the new version of Table 1 all clinical and laboratory variables (Covid-19 tests included) are now also shown for both disease subgroups.

  1. It would be nice to know the prevalence of IgG/IgM positivity in the subgroups.

               Reply to point 6: all patients with positive serology presented IgG positivity, no case of IgM positivity.

Please see the attachment for the revised manuscript.

Reviewer 3 Report

Cardiovascular manifestations in Multisystem Inflammatory Syndrome in Children associated with Covid-19 according to age

Multisystem inflammatory syndrome in children is a rare complication of Coronavirus disease 2019.  Twenty five children were included in this study who met the criteria for Covid -19 infections all were treated with IvIG, steroids and a few (8) with Anakarina in an effort to modify the immune response to the virus.  The children were divided into two groups: 6 years and older (12) and younger than 6 (13). The study is a single center observational study with the main outcome of a difference in cardiac involvement.  In younger children the coronaries were more involved and in older children the myocardium was more involved. The pathophysiology in younger children seems to be more like Kawasaki disease.  The main impact of the study is the good results of all 25 children who were all treated with immune modifying therapy.  There was no control group so the study is observational. 

The discussion did not include a hypothesis for the findings.  One possible hypothesis is that smaller children in the group less than 6 were easier to image; thus, the observed coronaries demonstrated more abnormalities.  Another possibility is the cells have a different tropism for the virus according to age.  Younger children have more cell markers inviting the virus in the coronaries than in the myocardium and older children have the opposite.  A third possibility is the immune system is age dependent, a reasonable assumption since the immune system learns from birth.

The paper should state the imaging modality of coronaries.  Why chose 6 other than it best represented the data.

My personal preference is not to use code for real words.  i.e.MISC-C for multisystem Inflammatory Syndrome in Children.  Abbreviation were introduced into medical literature to save money in typesetting.

The study is extremely important in demonstrating cardiac injury in this syndrome and the good outcome with the protocol chosen.  The discussion may be better by explaining the rationale for this protocol.

Author Response

Response to Reviewer 3

Multisystem inflammatory syndrome in children is a rare complication of Coronavirus disease 2019.  Twenty five children were included in this study who met the criteria for Covid -19 infections all were treated with IvIG, steroids and a few (8) with Anakarina in an effort to modify the immune response to the virus.  The children were divided into two groups: 6 years and older (12) and younger than 6 (13). The study is a single center observational study with the main outcome of a difference in cardiac involvement.  In younger children the coronaries were more involved and in older children the myocardium was more involved. The pathophysiology in younger children seems to be more like Kawasaki disease.  The main impact of the study is the good results of all 25 children who were all treated with immune modifying therapy.  There was no control group so the study is observational. 

The discussion did not include a hypothesis for the findings.  One possible hypothesis is that smaller children in the group less than 6 were easier to image; thus, the observed coronaries demonstrated more abnormalities.  Another possibility is the cells have a different tropism for the virus according to age.  Younger children have more cell markers inviting the virus in the coronaries than in the myocardium and older children have the opposite.  A third possibility is the immune system is age dependent, a reasonable assumption since the immune system learns from birth.

Response 1. Thank you for the very interesting comment. In the discussion we suggest that the prevalent Kawasaki-like presentation in younger subgroup could result from a different age-linked immune response to virus in genetically susceptible host. Although the exact etiopathogenesis of MIS-C and KD remains unclear, it’s hypothesized in both conditions an immune-mediated process triggered from virus infection. Even if the two entities present some overlapping clinical and immunological features, they develop a distinct inflammatory reaction. Maybe the immune response to Sars-Cov-2 is different in younger children in relation to a cross-reactive immunity to other coronaviruses. Moreover, it could be possible that the inflammatory reaction is less violent in pre-schoolage according to the immune system immaturity age-related, as you suggested. While, older patients with “typical” MIS-C presentation develop a severe hyperinflammatory reaction, maybe due to a silent inborn errors of immunity triggered by Sars-Cov-2, as described by Sancho-Shimizu et al. The resulting uncontrolled inflammatory response could be the main responsible for cardiomyocyte injury, often observed in this age-group. Another possibility is the cells could have a distinct tropism for the virus based on age.  Indeed, it’s described as Sars-Cov-2 receptors are widely expressed in organs with a possible different distribution, maturity and function in relation to children’s age. Younger children could have more cell markers inviting the virus in the coronaries than in the myocardium and older children could have the opposite. Further studies need to better explain our hypothesis.

  1. The paper should state the imaging modality of coronaries. 

Response 2.  Thanks for the comments. We described the imaging modality of coronaries in methods (line 66-71).

  1. Why chose 6 other than it best represented the data?

            Response 3. We choose the limit <6 years of age (0-5 years included) and ³6 years of age (6-21                         years included) because the Kawasaki Disease typical age is under 5 years, in consideration of the Kawasaki-like presentation in younger patients.

My personal preference is not to use code for real words.  i.e. MIS-C for multisystem Inflammatory Syndrome in Children.  Abbreviation were introduced into medical literature to save money in typesetting.The study is extremely important in demonstrating cardiac injury in this syndrome and the good outcome with the protocol chosen. 

  1. Reply: thank you very much for the kind appreciation of our work and for your interesting note on the use of abbreviations.  To this aim, we deleted the abbreviation CI (cardiac involvement). We will follow the indication of the Editor for the other, more common, abbreviations, such as MIS-C.

  • The discussion may be better by explaining the rationale for this protocol.

Response 4. Thanks for the interest to our protocol. We have revised the manuscript accordingly focusing on the rationale of the treatment protocol (line 268-272).

Please see the attachment for revised manuscript

Round 2

Reviewer 2 Report

The authors present a single-center retrospective observational study of 25 children with MIS-C. They described their population and focused on the cardiovascular manifestations in the subgroup < 6 years old compared to the subgroup > 6 years old. The main clinical message is that there is a significant difference in cardiac presentation of MIS-C patients int the preschoolers age versus the older with the first presenting with coronary arteries involvements and the latter with cardiac dysfunction. 

The authors significantly improved the manuscript in English language, syntax and most of all in the discussion. The addition of clinical information helps the readers to understand better. I have no further comments.